# Tolerance of *Pseudomonas* strain to the 2,4-D herbicide through a peroxidase system

**Elizangela Paz de Oliveira[1], Amanda Flávia da Silva Rovida[1], Juliane Gabriele Martins[2], Sônia Alvim Veiga Pileggi[2], Zelinda Schemczssen-Graeff[3], Marcos Pileggi[2]***

1 Departamento de Biotecnologia, Genética e Biologia Celular, Universidade Estadual de Maringá, Maringá, Brazil, 2 Departamento de Biologia Estrutural e Molecular e Genética, Laboratório de Microbiologia Ambiental, Setor de Ciências Biológicas e da Saúde, Universidade Estadual de Ponta Grossa, Ponta Grossa, Brazil, 3 Departamento de Biologia Celular, Universidade Federal do Paraná, Curitiba, Brazil

* mpileggi@uepg.br

**Data Availability Statement:** All relevant data are within the paper and its Supporting Information files.

## Abstract

Herbicides are widely used in agricultural practices for preventing the proliferation of weeds. Upon reaching soil and water, herbicides can harm nontarget organisms, such as bacteria, which need an efficient defense mechanism to tolerate stress induced by herbicides. 2,4-Dichlorophenoxyacetic acid (2,4-D) is a herbicide that exerts increased oxidative stress among bacterial communities. Bacterial isolates were obtained from the biofilm of tanks containing washing water from the packaging of different pesticides, including 2,4-D. The *Pseudomonas* sp. CMA-7.3 was selected because of its tolerance against 2,4-D toxicity, among several sensitive isolates from the biofilm collection. This study aimed to evaluate the antioxidative response system of the selected strain to 2,4-D. It was analyzed the activity of superoxide dismutase (SOD), catalase (CAT), ascorbate peroxidase (APX), and guaiacol peroxidase GPX enzymes, that are poorly known in the literature for bacterial systems. The *Pseudomonas* sp. CMA-7.3 presented an efficient response system in balancing the production of hydrogen peroxide, even at 25x the dose of 2,4-D used in agriculture. The antioxidative system was composed of Fe–SOD enzymes, less common than Mn–SOD in bacteria, and through the activities of KatA and KatB isoforms, working together with APX and GPX, having their activities coordinated possibly by quorum sensing molecules. The peroxide control is poorly documented for bacteria, and this work is unprecedented for *Pseudomonas* and 2,4-D. Not all bacteria harbor efficient response system to herbicides, therefore they could affect the diversity and functionality of microbiome in contaminated soils, thereby impacting agricultural production, environment sustainability and human health.

## 1 Introduction

The use of herbicides in agricultural practices combats the proliferation of undesirable weeds for the crop of interest and increase productivity. Because of the continued use of these pesticides, environmental problems are increasing and subsequently cause damage to organisms that are not the target of these herbicides [1].

**Funding:** The authors received no specific funding for this work.

**Competing interests:** The authors have declared that no competing interests exist.

Given the fight against weeds, 2,4-Dichlorophenoxyacetic acid (2,4-D) has become an ingredient of the most widely used active herbicides worldwide, as it exhibits a hormonal action upon an incorrect application. Even in small amounts, it can cause damage to culture sensitive to its active principle, 2,4-D is a systemic, pre- or post-emergent herbicide that is widely used in broadleaf crops, such as dicots, soybeans, rice, corn, and sugarcane [2].

The use of herbicides in agricultural practices causes environmental imbalances because these herbicides have electronegative elements in their chemical structure. The 2,4-D molecule comprises chlorine (Cl) and hydroxyl radical (OH), which can be reactive and harmful to various biological molecules, such as genetic material, proteins, and membrane lipids [3].

Herbicides present in soil can move to aquatic ecosystems by leaching and negatively affect organisms in these environments [4].

Humans are exposed to the 2,4-D herbicide, as are 27–28% of Canadian farmers working with corn, soybeans, and other grains [5]. Traces of pesticides, including 2,4-D, were found in meat products in markets in Oman [6]. This herbicide was found in high concentrations in aquatic environments in The Kibale National Park in Uganda, with water samples inducing thyroid and estrogen axis disrupting *in vivo* activities in *Xenopus laevis*, plus affecting development and behavior of this model organism [7]. 2,4-D is a pesticide that can be found in Sinos River, Brazil, which showed cytotoxicity and genotoxicity to HEp-2 epithelioid-type cell line [8]. These data show the possible harmful effects of 2,4-D for humans, who have high exposure to this herbicide [9–11].

The presence of herbicides can affect microorganisms, resulting in the loss of their ecological functions in the loss of microbial diversity [12]. Many microorganisms help in maintaining soil fertility, nutrient cycling, and nitrogen fixation. Due to a decrease in microbial diversity in the soil, effective strategies are needed to adapt to these conditions and maintain ecological functionality [13].

Upon exposure to herbicides, microorganisms can produce reactive oxygen species (ROS) and induce oxidative stress [14]. ROS are produced by different physiological systems, such as aerobic metabolism, and are important in cell signaling pathways. A high concentration of ROS can exert harmful effects on cellular components, such as lipids, proteins, and nucleic acids [15].

Response systems comprising antioxidative enzymes, such as superoxide dismutase (SOD), catalase (CAT), and peroxidase, maintaining ROS at nontoxic concentrations, while coordinating the balance between their production and degradation [14]. Different studies have demonstrated that the interference of CAT activity in the tolerance of bacteria against xenobiotics, such as a strain of *Escherichia coli*, *Pantoea ananatis* and *Bacillus megaterium* exhibited an increased CAT activity in response to the herbicide mesotrione [13, 16, 17]. The enzymes APX and GPX in a *Pseudomonas* strain also showed a significant increase in the mid-log phase, evidencing the cooperative action with CAT for the control of $H_2O_2$ [18].

Therefore, this study aimed to evaluate the antioxidative response system of the *Pseudomonas* sp. CMA-7.3, which was isolated from the biofilm present in a tank containing washing water from the packaging of different pesticides, including 2,4-D. This system comprises little studied enzymes such as SOD, CAT, APX, and GPX in bacterial systems.

## 2 Material and methods

### 2.1 Selected strain

The bacterium *Pseudomonas* sp. CMA-7.3 was obtained from the Collection of Environmental Microorganisms of the Laboratory of Environmental Microbiology at the State University of

Ponta Grossa, Brazil. It was isolated from water that was used for washing herbicide containers and kept in 30% glycerol at −80˚C [19].

## 2.2 Tolerance test

*Pseudomonas* sp. CMA-7.3 was seeded in Petri dishes containing Agar Luria Bertani (LA) (10 g/L tryptone, 5 g/L yeast extract, 10 g/L NaCl, and 20 g/L agar) and treated with 10x (14.4 mM) and 25x (36 mM) herbicide concentrations, which were compared with the control treatment. The plates were incubated for 24 h at 30˚C and the isolates that showed growth were considered as tolerant.

## 2.3 Bacterial growth conditions

*Pseudomonas* sp. CMA-7.3 was grown in Luria Bertani Broth medium (LB: 10 g/L tryptone, 5 g/L yeast extract, 10 g/L NaCl) and treated with 10x (14.4 mM) and 25x (36 mM) herbicide concentrations. These treatments were compared with the control, which contained only the culture medium and bacteria. The bacteria grew at 30˚C with agitation of 120 rpm. The tests were conducted in triplicates in 250-ml flasks containing 100 mL LB. The inoculants were standardized to start at an optical density (OD) of 0.05, and the bacterial growth was evaluated in a spectrophotometer at an absorbance of 600 nm. The samples were diluted upon reaching approximately 1.0 values, and the values were multiplied by the corresponding dilution factors.

## 2.4 Conventional PCR reactions for the amplification of ribosomal 16S gene

For PCR, 30 ng template DNA was used. For amplification, primer forward fD1 (5′–CCGAATTCGTCGACAACAGAGTTTGATCCTGGCTCAG–3′) and rD1 (5′–CCCGGGATC–CAAGCTTAAGGAGGTGATCCAGCC–3′) were used. For amplification, 5 µL Pfu DNA polymerase buffer, 0.5 U Pfu DNA polymerase, 0.2 µM dNTP mix, 0.4 µM forward and reverse primers, and sterile water were used for the final reaction volume of 20 µL. The reaction was conducted in a thermocycler under the following conditions: Initial Denaturation at 95 µC/2 min; Denaturation at 95 µC/30 s; Annealing at 57.7 µC/30 s; Extension at 72 µC/3 min (40 cycles); Final-Extension at 72 µC/5 min & 4 µC/∞.

The PCR product was analyzed using electrophoresis on 1.5% agarose gel with 0.5 µg/mL ethidium bromide in TAE buffer (40 mM Tris base and 20 mM acetate/1 mM of ethylenediaminetetraacetic acid [EDTA]). The electrophoretic run was performed in a horizontal bowl at 5 V/cm of gel. The gel was visualized, and the image was registered with the help of a Chemidoc–XRS image analysis device and Quantity One–SW software (BioRad).

## 2.5 Extraction of agarose gel fragments

The agarose gel was cut with a scalpel at the location of the corresponding bands. The gel fragments were placed in microtubes, and the DNA was extracted with the aid of a specific kit for DNA extraction in gel (Gel Band Purification kit, GE®).

## 2.6 Sequencing reaction and analysis

The sequencing protocol BigDye® Terminator v3.1 Cycle Sequencing Kit (Applied Biosystems) was used, with a total reaction volume of 10 µL. Of this 10 µL, 3 µL was the Big Dye reagent and the remaining 7 µL was the mixture of template DNA, oligonucleotide, and sterile water. The sample was subjected to two sequencing reactions: one using the forward fD1

primer (5′–CCGAATTCGTCGACAACAGAGTTTGATCCTGGCTCAG–3′) and the other using the reverse primer rD1 (5′–CCCGGGATCCAAGCTTAAGGAGGTGATCCAGCC–3′). These reactions were conducted in a conventional thermocycler at the following conditions: Denaturation at 95˚C/20 min; Annealing at 50˚C/15 s; Extension at 60˚C/4 min (35 cycles) & 4˚C/ ∞. For precipitation of the product to be sequenced, sodium acetate/EDTA (1.5 M/0.25 M) (1/10th of the initial reaction volume) was added, followed by chilled absolute ethanol (thrice the reaction volume); this mixture was homogenized well and incubated on ice for 10 min. Then, it was centrifuged at 4˚C for 20 min at 20,000× g. The supernatant was removed, and the pellet was washed with cold 70% ethanol (500 μL) by passing the liquid through the pellet without homogenizing. It was then centrifuged at 4˚C for 10 min at 20,000× g. The supernatant was removed, and the pellet was dried at room temperature within the laminar flow for 15 min for completely drying the ethanol that can interfere with sequencing.

The product was resuspended in 10 μL formamide. Sequencing was performed in a 3500xL Genetic Analyzer sequencer by capillary electrophoresis (Applied Biosystems® 3500) and the results were analyzed using Chromas Pro version 1.5 software. The alignment and analysis of the acquired sequences were conducted using Clustal (http://www.ebi.ac.uk/clustalw/). Later, the most representative sequence chosen was analyzed against the available database using the nucleotide blast tool BLAST (http://blast.ncbi.nlm.nih.gov/Blast.cgi). The string was deposited in the GenBank database of the National Center for Biotechnology Information (NCBI) under the code MW 766917.1.

## 2.7 Phylogenetic tree

The phylogenetic tree was constructed for comparing the sequence of the CMA-7.3 strain with sequences deposited in the NCBI. For analysis, 10 nucleotide sequences were used, in which 7 were similar and 3 were chosen randomly, which did not show similarity with the CMA-7.3 strain. The evolutionary history was inferred using the neighbor-joining method. The tree was drawn to scale, with the lengths of the branches in the same units as the evolutionary distances. These distances were calculated using the maximum compound likelihood method. The codon positions included were 1st + 2nd + 3rd + and noncoding. All ambiguous positions were removed for each pair of strings. Evolutionary analyses were conducted using the Molecular Evolutionary Genetics Analysis (MEGA) version X software.

## 2.8 Composition of 2,4-D herbicide

2,4-dichlorophenoxyacetate (2,4-D dimethylamine) comprises 806 g/L of the active molecule of herbicide, equivalent to 2,4-D acid corresponding to 670 g/L, and 429 g/L of other ingredients (42.9% w/v) [20].

## 2.9 Bacterial growth curve

The strain was grown as previously described in this study. After this period, the culture was inoculated in the following concentrations of herbicide: 0x, 10x, and 25x. The inoculants were standardized to begin with an OD of 0.05 and absorbance of 600 nm. The samples were diluted upon reaching the OD values of approximately 1.0, and the values were multiplied by the corresponding dilution factors.

## 2.10 Cell viability

Bacterial cultures were obtained under culture conditions with treatments with 2,4-D as previously described in this study. The cells were recovered by centrifugation and diluted in 0.9%

NaCl buffer for removing the residues of herbicides. The cultures, in triplicates, were incubated in LA plates. Dilutions were made until 25–300 colonies were obtained per plate, after incubation at 30 μC for 24 h [13].

## 2.11 Protein extraction for oxidative stress analysis

Bacterial cultures were grown as previously described in this study. The proteins were extracted in three periods: mid-log phase, late-log phase, and stationary phase. The culture was centrifuged at 5,000 g for 15 min, and the precipitate was macerated with liquid nitrogen and homogenized in 1:10 m/v of a 100 mM solution of potassium phosphate buffer (14,520 g/L $K_2HPO_4$ and 2,260 g/L $KH_2PO_4$; pH 7.5), 0.372 g/L EDTA, 0.462 g/L DL-dithiothreitol, 5% (w/w) polyvinyl polyvinylpyrrolidone (10:1 buffer volume: sample weight) at 4 μC. The mixture was then centrifuged at 10,000 g for 30 min. The supernatant was stored at −80 μC. Protein concentrations were measured using [21] method, with bovine serum albumin as a standard. The results were expressed in μmol protein/g of fresh weight.

## 2.12 Hydrogen peroxide

The quantification of hydrogen peroxide ($H_2O_2$) was conducted by reacting 200 μL sample (100 mg protein extract homogenized with 1 ml 0.1% trichloroacetic acid [TCA] and centrifuged at 10,000 g for 15 min at 4 μC) with 200 μL potassium phosphate buffer (pH: 7.5) and 800 μL 1 M potassium iodide for 1 h on ice in the dark. The iodine, released in this reaction, was quantified in a spectrophotometer at 390 nm. The results were expressed in μmol/g of fresh mass [22].

## 2.13 Lipid peroxidation

Bacterial growth and pre-inoculation were performed as previously described in this study. Lipid peroxidation was determined in a spectrophotometer at 530 and 600 nm by measuring the amount of malondialdehyde (MDA) produced, which is a metabolite reactive to 2-thiobarbituric acid (TBA). To a volume of 250 μL of the sample (described previously in this study), 200 μL potassium phosphate buffer (pH: 7.5) and 1 mL of 20% TCA + 0.1% TBA was added, which was maintained for 30 min in a water bath at 97 μC. The sample remained in the ice for 10 min before centrifugation at 10,000 g for 10 min and the amount of MDA in the supernatant was estimated using a spectrophotometer. The amount of MDA was calculated using an extinction coefficient of 155 mM cm$^{-1}$. The amount of MDA was expressed as μmol MDA g$^{-1}$ of fresh weight.

## 2.14 Isoforms (SOD)

Superoxide dismutase (SOD) isoforms were separated using 12% nondenaturing polyacrylamide electrophoresis (PAGE) gels [13]. The gels were divided vertically into three parts and kept in the dark. The first part was immersed in 100 mM potassium phosphate buffer (pH: 7.8), the second part was immersed in 100 ml of 100 mM potassium phosphate buffer containing 2 mM KCN and 1 mM EDTA, and the third part was immersed in 100 mL of 100 mM potassium phosphate buffer with 5 mM $H_2O_2$ and 1 mM EDTA. Isoforms were classified as Mn–SOD if resistant to both inhibitors (KCN and $H_2O_2$), as Fe–SOD if resistant to KCN and inhibited by $H_2O_2$, and as Cu/Zn–SOD if inhibited by both substances [23].

## 2.15 SOD activity in nondenaturing PAGE

Electrophoresis was performed on 12% polyacrylamide separation gels and 4% polyacrylamide packaging gel at a current of 15 mA for 3 h using 20 µg of each protein extract per channel as previously described in this study. The gels were washed with deionized water and incubated in the dark at room temperature in 50 mM potassium phosphate buffer (pH: 7.8) comprising 1 mM EDTA, 0.05 mM riboflavin, 0.1 mM tetrazolium nitroblue (NBT), and 0,3% N, N, N′, N′-tetramethylethylenediamine (TEMED). This solution was discarded after 30 min of reaction. The gels were washed with deionized water and placed under fluorescent lighting to identify the bands.

## 2.16 Catalase activity in nondenaturing PAGE

CAT activity was determined using nondenaturing PAGE in 12% polyacrylamide separation gels and 4% packaging gel, as reported by [14]. A current of 15 mA per gel was applied for 17 h at 4 µC with 15 µg of protein from the samples described previously in this study. The gels were washed with deionized water (3 times for 15 min) and incubated in 0.003% $H_2O_2$ for 10 min and transferred to a 1% (w/v) FeCl3 solution and 1% K3Fe (CN$_6$) solution (w/v) for 10 min for developing of bands.

## 2.17 Ascorbate Peroxidase (APX) activity

APX activity was determined in a reaction mixture using 650 µL potassium phosphate buffer (80 mM; pH: 7.0), 100 µL EDTA (1 mM), 100 µL ascorbic acid (5 mM), and 100 µL $H_2O_2$ (1 mM) and quantified in a spectrophotometer at 25 µC. The reaction was started with the addition of 50 µL protein extract as previously described in the present study and APX activity was determined following the decomposition of $H_2O_2$ at 290 nm for 1 min. The activity was expressed in µmol/min/mg protein [24].

## 2.18 Guaiacol Peroxidase (GPX) activity

GPX activity was determined using 390 µL citrate buffer (0.2 M dibasic disodium phosphate and 0.1 M citric acid, pH: 5.0), 25 µL guaiacol, and 25 µL $H_2O_2$ (3%). These constituents were homogenized with 25 µL of protein extract as described in Section 2.11 and incubated in a water bath at 30˚C for 15 min. Then, 25 µL sodium metabisulfite (2%) was added and the GPX activity was measured in a spectrophotometer at 450 nm. The activity was expressed in µmol/min/mg protein [25].

## 2.19 Statistical analysis

Statistical analyses of the tests were performed using analysis of variance (two-way ANOVA), followed by Tukey's post hoc test. Significance was set at $p < 0.05$ using the GraphPad Prism 6 program (GraphPad Software, San Diego, CA, USA) (S1 Appendix).

## 3 Results and discussion

### 3.1 Tolerance indicators

**3.1.1 Bacterial tolerance against 2,4-D.**   From a tank of water used to wash containers of different pesticides, including the herbicide 2,4-D, 33 bacterial strains were isolated from a biofilm, 12 of which were considered sensitive and 21 were tolerant up to 10x concentration of 2,4-D. From 21 bacterial strains that were tolerant up to 10x 2,4-D concentration, 14 of those were also tolerant up to 25x, including the *Pseudomonas* sp. CMA-7.3. The phylogeny of this

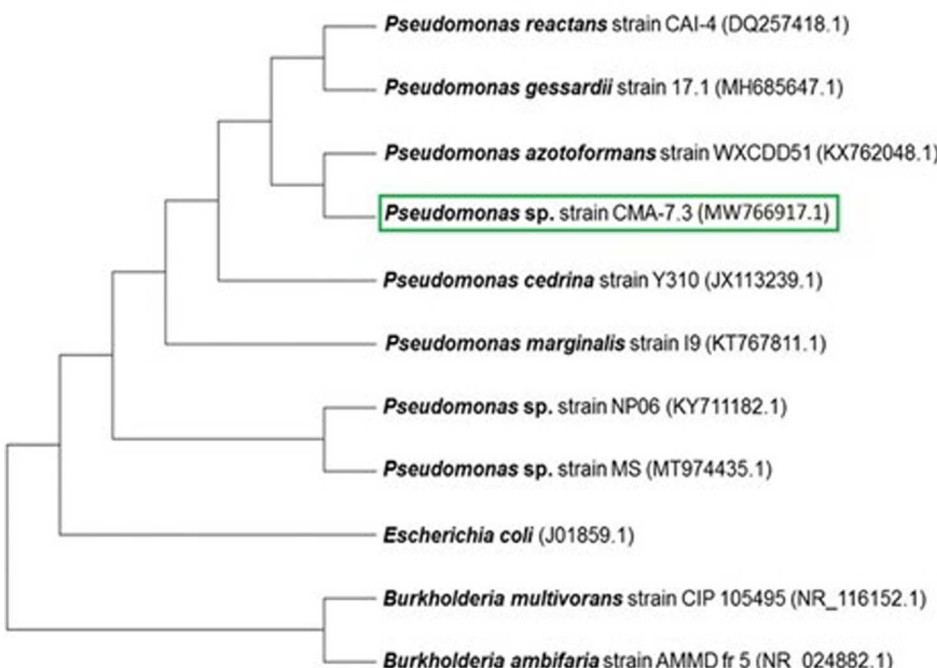

**Fig 1. Phylogenetic tree comparing the sequence of the CMA-7.3 strain with sequences deposited in the NCBI.** For analysis, 10 nucleotide sequences were used, in which 7 were similar and 3 were chosen randomly, which did not show similarity with the CMA-7.3 strain. The evolutionary history was inferred using the neighbor-joining method. The tree was drawn to scale, with the lengths of the branches in the same units as the evolutionary distances. These distances were calculated using the maximum compound likelihood method. The codon positions included were 1st + 2nd + 3rd + and noncoding. All ambiguous positions were removed for each pair of strings (option of exclusion in pairs). Evolutionary analyses were performed using the MEGA version X software.

strain with other sequences of the 16S ribosomal RNA gene is shown in the phylogenetic tree (Fig 1). Because some isolates could not grow in a medium containing 2,4-D, this herbicide was considered toxic and, therefore, capable of exerting selective pressure on bacterial cells [19].

Although 2,4-D influences the structure of microbiomes in soil, even with a low half-life [26], there are few studies that describe the effects of 2,4-D on bacteria of the genus *Pseudomonas*, reporting degradation processes, without describing bacterial systems of tolerance to the herbicide [27]. One exception is the description of the role of quorum sensing signaling molecules in the 2,4-D response to 2,4-D [28].

**3.1.2 Growth curve.** In the growth curve of *Pseudomonas* sp. CMA-7.3, the 0x concentration was used as a basis to determine the mid-log phase (11 h of incubation), late-log phase (14 h), and stationary phase (17 h) (Fig 2), during which the stress data and response systems were obtained. At the 0x concentration, the line shows greater exponential growth compared with 10x and 25x concentrations, mainly in the stationary phase, despite considerable rates in these concentrations, characterizing this line as tolerant.

The growth kinetics of bacterial strains in the presence of herbicides, in addition to the degradation kinetics, are considered indicators of degradation models, including the presence of toxic intermediates [29]. In our work, growth kinetics were used to characterize the stress response system generated by 2,4-D at different times of the growth phases.

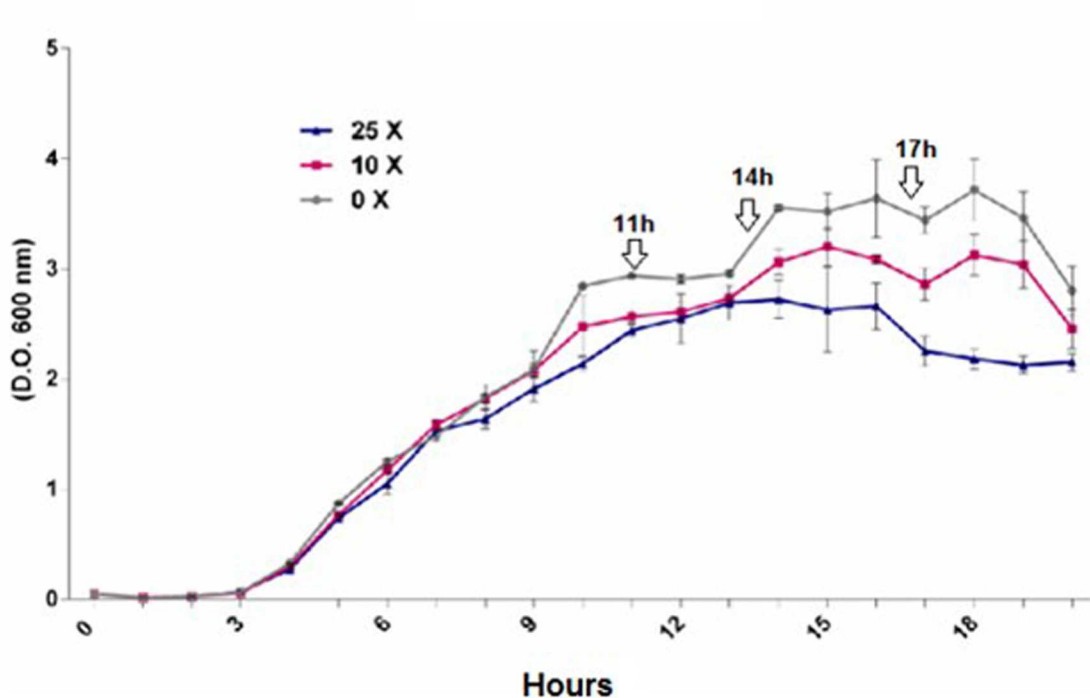

**Fig 2. Growth curve of *Pseudomonas* sp. CMA 7.3 undergoing treatments with 0x, 1x, and 25x concentrations.** Incubation times of 11 h, 14 h, and 17 h were stipulated for the mid-log, late-log, and stationary phases, respectively. Readings were taken at 600 nm.

### 3.2 Stress indicators

**3.2.1 Cell viability.** Cell viability corresponds to the number of cells with the capacity to perform cell division at certain incubation times and culture conditions. The viability of *Pseudomonas* sp. CMA-7.3 showed no significant differences between treatments and growth phases, thereby indicating that a response system is responsible for tolerance, despite the indication that 2,4-D is toxic at higher concentrations, as observed by the decrease in viability values at 25x concentration in the stationary phase (Fig 3).

Bacterial cells have complex mechanisms, which can increase their survival potential after meeting various stress conditions [18]. Antioxidant enzymes play a fundamental role in these situations; however, other systems can play an important role in the survival of bacterial populations against a xenobiotic agent [30]. Strains that have a metabolic response system favoring their survival in a toxic environment can maintain their cellular integrity, as they can manage to balance the production of $H_2O_2$ in the cell, while protecting themselves against toxic effects [31]. However, the antioxidant enzyme system is little explored in these organisms.

**3.2.2 $H_2O_2$ quantification.** The results of the $H_2O_2$ quantification of *Pseudomonas* sp. CMA-7.3 are shown in Fig 4. The mid-log phase is characterized by intense growth (Fig 2) and metabolism. Thus, the amounts of $H_2O_2$ are significantly higher, but they decrease progressively until the stationary phase, in all treatments, thereby indicating the functioning of the antioxidative system of this strain. However, the drop in the amount of $H_2O_2$ is more significant at 0x concentration than at 25x concentration, which suggests that the toxicity of 2,4-D at

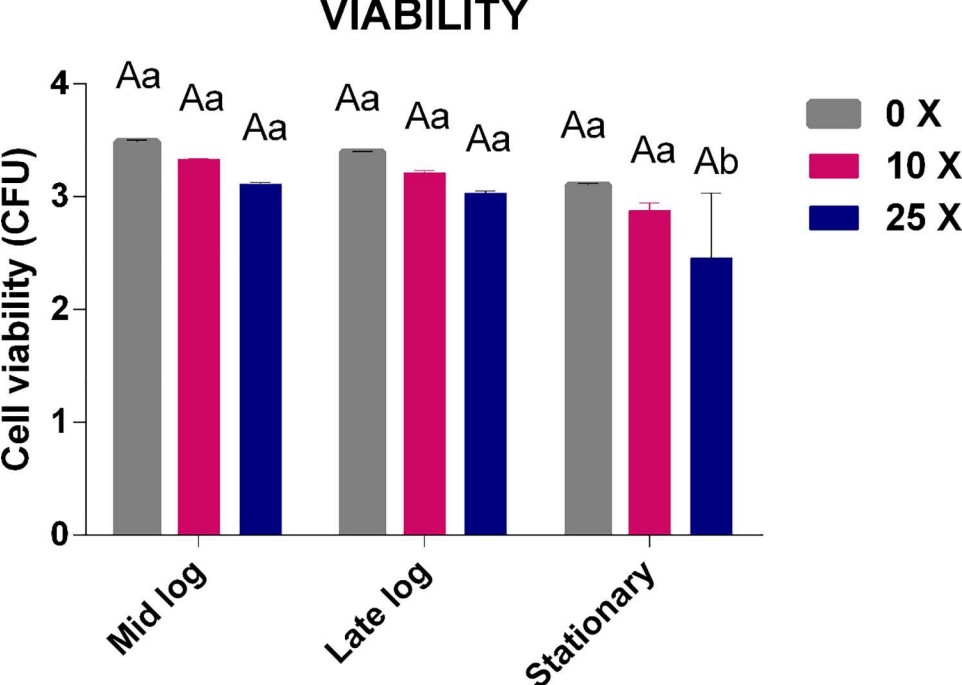

**Fig 3. Cell viability of *Pseudomonas* sp. CMA 7.3 undergoing treatments with 0x, 1x, and 25x concentrations in the mid-log, late-log, and stationary phases.** The data were obtained in triplicate for each treatment and statistically analyzed using the complete block design through the analysis of variance (two-way ANOVA), followed by Tukey's post hoc test. Error bars represent statistically significant differences between treatments at the same time. Capital letters represent statistically significant differences between treatments at different times. Significance was set at $p < 0.05$.

higher concentrations interferes negatively in the response system of *Pseudomonas* sp. CMA-7.3.

One of the toxic effects of 2,4-D is the oxidative stress produced by the increase in ROS [32], thus defense mechanisms are important to maintain the balance of $H_2O_2$ as well as enzyme activities. $H_2O_2$ is stable in abiotic environment, at room temperature, and in neutral pH conditions, and it quickly kills any type of cell by producing highly reactive hydroxyl radicals [33].

Enzymes such as SOD and CAT help in regulating ROS to avoid cell damage, therefore [13] these characteristics have been associated with survival in stressful environments [34], as well as the activities of the enzymes APX and GPX, still little explored in the control of $H_2O_2$ induced by herbicides in bacteria [18].

**3.2.3 Quantification of Malondialdehyde (MDA).** MDA is an indicator of oxidative stress in adverse environmental conditions. It is a toxic aldehyde that is released when an ROS reacts with unsaturated lipids of the cell membrane, thereby causing lipid peroxidation [35]. Lipids are responsible for maintaining the integrity of cell membranes. Lipid peroxidation exerts a toxic effect when it increases peroxide and MDA production, thus changing the structure, composition, and dynamics of membranes. The ROS linked to peroxidation that we studied was $H_2O_2$. As a result, this peroxidation in addition to affecting the membrane structure can disrupt other molecules, such as DNA and proteins [36].

The MDA quantifications (Fig 5) showed that there were no significant differences at the 0x, 10x, and 25x concentrations in the mid-log and late-log phases, thereby showing the efficient control of this stress indicator, even in the presence of high concentrations of 2,4-D. An

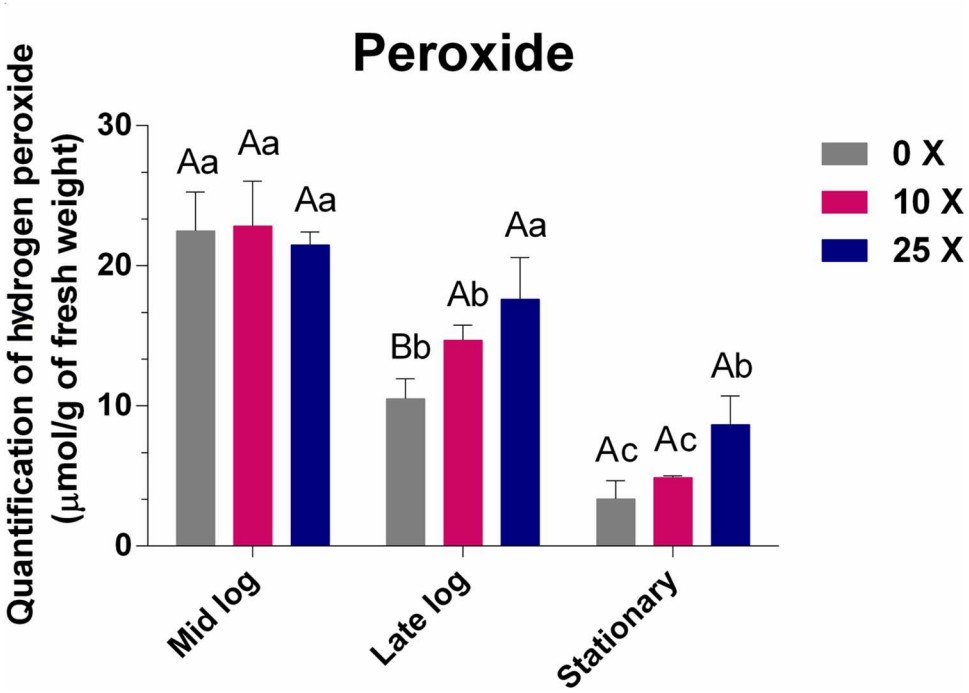

**Fig 4. Quantification of peroxide in *Pseudomonas* sp. CMA 7.3 in treatments with 0x, 1x, and 25x concentrations in the mid-log, late-log, and stationary phases.** The data were obtained in triplicate for each treatment and analyzed statistically using the complete block design through the analysis of variance (two-way ANOVA), followed by Tukey's post hoc test. Error bars represent statistically significant differences between treatments at the same time. Capital letters represent statistically significant differences between treatments at different times. Significance was set at $p < 0.05$.

approximate correspondence between levels of peroxide (Fig 4) and MDA (Fig 5) was found only up to the mid-log phase, which could represent lipid peroxidation.

Oxidative stress is caused by the production of uncontrolled $H_2O_2$ in cells that induce lipid peroxidation, thereby causing damage to membrane fatty acids [37]. In *E. coli*, it was observed that after exposure to 2,4-D, cell growth is inhibited, further suggesting that this herbicide interrupts cell division facilitated by membrane damage [38].

However, the correspondence between peroxide and MDA, as already noted, does not occur in the late-log and stationary phases of *Pseudomonas* sp. CMA-7.3, making the hypothesis of lipid peroxidation proportional to peroxide unfeasible. Different bacterial species show changes in their lipid composition in response to the toxicity of different herbicides, such as *E. coli* receiving treatment with gramoxone [22] and *Pantoea ananatis* treated with mesotrione [17]. Thus, it is possible that changes in the composition of membrane fatty acid residues, throughout the growth phases, may decrease the production of MDA and lipid peroxidation, as a response system to herbicides, through membrane stabilization [17].

### 3.3 Response enzymatic system

**3.3.1 SOD enzyme activity.** Antioxidative enzymes are important for controlling the amount of ROS in both stress-free and herbicide-induced metabolism conditions, without which there may be high levels of damage to the cell membranes, affecting bacterial viability.

SOD is considered the first enzyme that acts in the cellular defense against ROS because it catalyzes the dismutation of superoxide into $O_2$ and $H_2O_2$. SOD is classified according to its

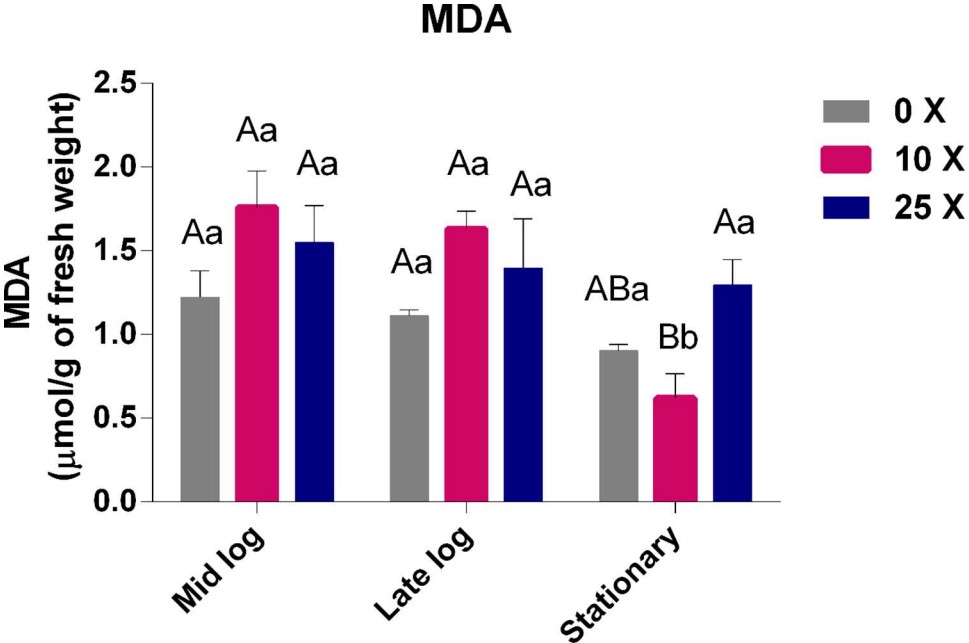

**Fig 5. Quantification of MDA in *Pseudomonas* sp. CMA 7.3 in treatments with 0x, 1x, and 25x concentrations in the mid-log, late-log, and stationary phases.** The data were obtained in triplicate for each treatment and analyzed statistically using the complete block design through the analysis of variance (two-way ANOVA), followed by Tukey's post hoc test. Error bars represent statistically significant differences between treatments at the same time. Capital letters represent statistically significant differences between treatments at different times. Significance was set at $p < 0.05$.

metallic cofactor: manganese (Mn), Iron (Fe), and copper–zinc (Cu–Zn). These enzymes are found in several organisms, which can use one or more types of isoenzymes [39]. Fe–SOD was the isoenzyme identified in *Pseudomonas* sp. CMA-7.3 (Fig 6), being active in the three phases evaluated, with more intensity in the stationary phase. Fe–SOD is preferentially expressed under conditions of high concentrations of extracellular iron. At low concentrations of extracellular iron, Mn–SOD is expressed. Thus, the culture medium can induce differential expression of these isoenzymes. For example, LB medium induces the expression of Fe–SOD in *P. aeruginosa* [39]. The same medium was used with *Pseudomonas* sp. CMA-7.3.

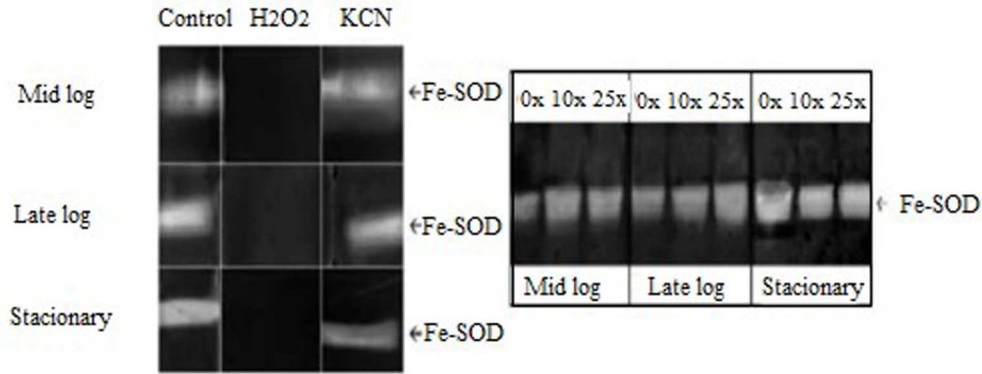

**Fig 6. Characterization of SOD isoforms in PAGE, which were obtained from the extracts of *Pseudomonas* sp. CMA 7.3 grown in LB, treated with KCN and $H_2O_2$, in the mid-log, late-log, and stationary incubation phases.**

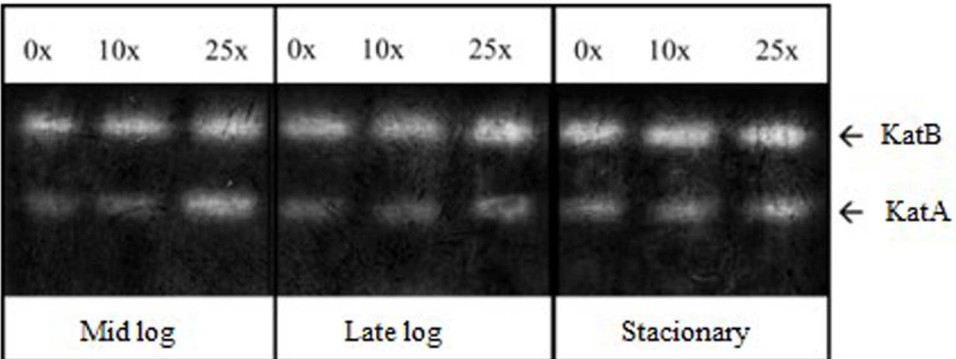

**Fig 7. Characterization of CAT isoforms in PAGE, which were obtained from the extracts of *Pseudomonas* sp. CMA 7.3 undergoing treatments with 0x, 1x, and 25x concentrations of 2,4-D from samples obtained in the mid-log, late-log, and stationary phases.**

**3.3.2 CAT enzyme activity.** CAT is an enzyme that acts in the conversion process of $H_2O_2$, a product dismuted by SOD, converting it into $O_2$ and $H_2O$ [40]. This enzyme has two isoforms: KatA and KatB (Fig 7). KatA, which is active in all growth phases and more related to the control of high concentrations of $H_2O_2$ and KatB, which is active only in the presence of $H_2O_2$ in *Pseudomonas aeruginosa* [41]. KatB provides resistance to exogenous hydrogen peroxide in *P. aeruginosa* [41–43]. For *Pseudomonas* sp. CMA 7.3, KatB has a more intense expression than KatA, with the highest activity being achieved in the stationary phase. KatA will present its most intense activities only in treatments with 25x concentration of 2,4-D, thus characterizing the requirement of $H_2O_2$ control in this concentration.

Strains of *Pseudomonas aeruginosa* isolated from biofilms could withstand the stress induced by $H_2O_2$ by regulating the activities of enzymes Mn-SOD, Fe-SOD, and KatA through quorum sensing signaling molecules [44]. This mechanism could explain the different levels of SOD (Fig 6) and CAT (Fig 7) activities throughout the three growth phases (Fig 2).

**3.3.3 Quantification of APX enzyme activity.** APX are peroxidases that metabolize $H_2O_2$ to $H_2O$ and $O_2$ in plant cells, using ascorbate as an electron donor. APX is one of the main regulatory enzymes for ROS [45, 46]. APX activity in *Pseudomonas* sp. CMA-7.3 showed an increase during the growth phases, being more significant at 25x concentration than at 0x concentration (Fig 8).

**3.3.4 Quantification of GPX enzyme activity.** The GPX enzyme, in the same way as CAT and APX enzymes, is important in the conversion of $H_2O_2$ into $H_2O$, using guaiacol as an electron donor. As with the other enzymes studied in this work, the GPX enzyme showed an activity increase over the growth phases, with a higher significance at 25x concentration in the stationary phase, thereby assisting in the control of $H_2O_2$ rates (Fig 9).

Reports have shown that the activities of APX and GPX enzymes are better known in plants than in bacteria. The closest reports in bacteria are those about symbiosis with plants. For example, when *B. cepacia* encountered different concentrations of the herbicide glyphosate, a decrease in the activities of CAT, APX, and GPX was observed [35]. The authors only concluded that the bacteria could be suppressing the activity of plant enzymes; they did not describe the activity of the bacterial enzymes.

Some bacterial strains, in the presence of herbicides, managed to maintain the balance of $H_2O_2$ with the aid of antioxidant enzymes. *Escherichia coli* K-12, a nonenvironmental strain, was found to survive in the presence of gramoxone, even without prior contact with this herbicide, implying this strain is a model of phenotypic plasticity for adaptation to this herbicide

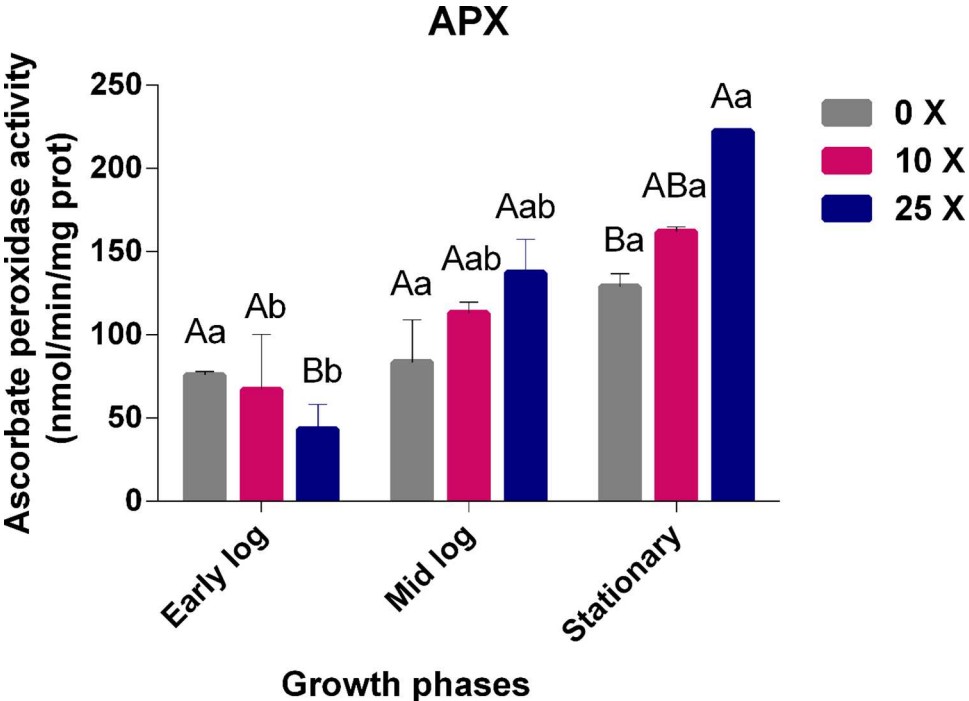

**Fig 8. Quantification of APX in *Pseudomonas* sp. CMA-7.3 undergoing treatments with 0x, 10x, and 25x concentrations of 2,4-D in the mid-log, late-log, and stationary phases.** The data were obtained in triplicate for each treatment and analyzed statistically using the complete block design through the analysis of variance (two-way ANOVA), followed by Tukey's post hoc test. Error bars represent statistically significant differences between treatments at the same time. Capital letters represent statistically significant differences between treatments at different times. Significance was set at $p < 0.05$.

[22]. Similar mechanisms of adaptation behavior were observed in *Pseudomonas* sp. CMA-7.3 strain, but as an environmental strain having previous contact with the herbicide 2,4-D. The enzymatic control of $H_2O_2$ is poorly documented for bacteria, and this work is unprecedented for *Pseudomonas* and 2,4-D, firstly describing the enzymatic role of APX and GPX in assisting CAT in $H_2O_2$ control in response to herbicides in bacteria. Since there are reports of coordinated gene regulation between SOD and CAT isoforms in *Pseudomonas* through quorum sensing signaling molecules [44], and the types and number of these molecules depend on the bacterial density and the presence of herbicides, including 2,4-D [28], it is possible that the quorum sensing is also related to the coordinated activities of CAT, APX, and GPX in *Pseudomonas* sp. CMA-7.3.

## 4 Conclusions

*Pseudomonas* sp. CMA-7.3 was isolated from an environment with different pesticides, including 2,4-D. This herbicide is considered toxic for this strain because of the decrease in growth rates and viability with an increase in the concentration of this herbicide. In the response enzymatic system, only Fe–SOD enzymes, less common than Mn–SOD, were detected. It may be possible that due to the isolation and cultivation environment conditions, their activity in the stationary phase increased to control the ROS superoxide induced in this phase. MDA concentrations, indicating lipid peroxidation, were also controlled, but possibly more by structural systems, such as changes in the lipid composition, according to literature data. This enzyme response system, possibly associated with structural changes, is important for the survival of

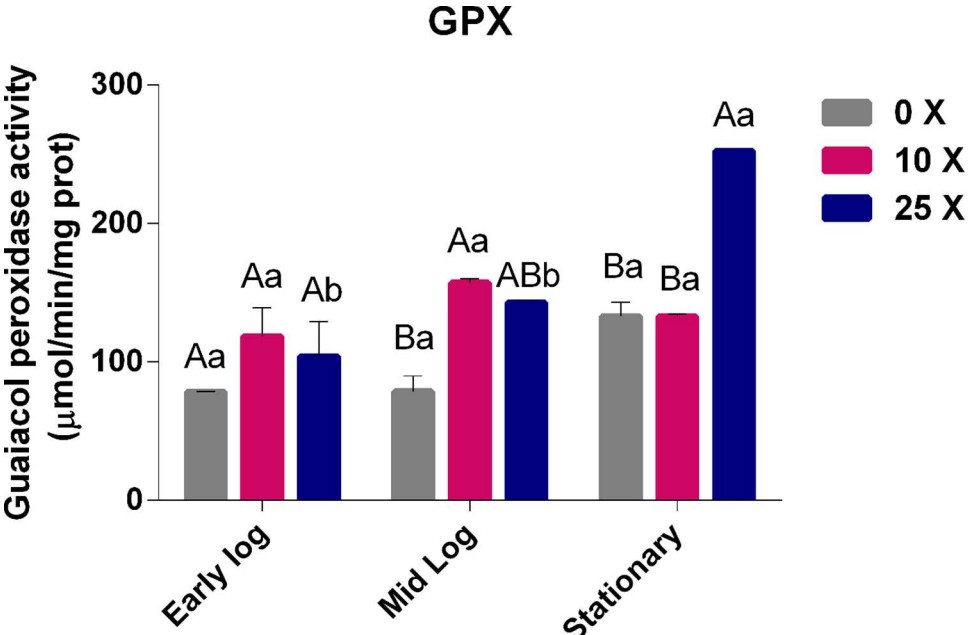

**Fig 9. Quantification of GPX in *Pseudomonas* sp. CMA-7.3 undergoing treatments with 0x, 10x, and 25x concentrations of 2,4-D in the mid-log, late-log, and stationary phases.** The data were obtained in triplicate for each treatment and analyzed statistically using the complete block design through the analysis of variance (two-way ANOVA), followed by Tukey's post hoc test. Error bars represent statistically significant differences between treatments at the same time and capital letters represent statistically significant differences between treatments at different times. Significance was set at $p < 0.05$.

strains such as *Pseudomonas* sp. CMA-7.3 in environments contaminated with toxic substances. The most striking in this work were the fact this strain showed an efficient response system to control the amount of peroxide. This tolerance was related to the activities of CAT enzymes and their isoforms, namely, KatA and KatB, working together with APX and GPX, having their activities coordinated possibly by quorum sensing molecules. This peroxide control is poorly documented for bacteria, and this work is unprecedented for *Pseudomonas* and 2,4-D. Probably, not all bacteria harbor an efficient response system to herbicides: approximately 36% of the isolates obtained from the water used to wash pesticide packaging failed to grow in the 1x concentration of 2,4-D. Therefore, this herbicide affects the population structure of bacteria in contaminated soils and interferes in the diversity and functionality of microbiomes, thereby impacting agricultural production, environmental sustainability, and human health.

## Supporting information

**S1 Appendix. Statistical analysis.**
(PDF)

**S1 Raw images.**
(DOCX)

## Acknowledgments

The authors want to thank Maria Janina Pinheiro Diniz for assisting in the preparation and execution of the experiments.

## Author Contributions

**Conceptualization:** Elizangela Paz de Oliveira, Marcos Pileggi.

**Data curation:** Elizangela Paz de Oliveira, Amanda Flávia da Silva Rovida, Juliane Gabriele Martins, Zelinda Schemczssen-Graeff, Marcos Pileggi.

**Formal analysis:** Elizangela Paz de Oliveira, Juliane Gabriele Martins, Sônia Alvim Veiga Pileggi, Zelinda Schemczssen-Graeff, Marcos Pileggi.

**Funding acquisition:** Sônia Alvim Veiga Pileggi, Marcos Pileggi.

**Investigation:** Elizangela Paz de Oliveira, Sônia Alvim Veiga Pileggi, Marcos Pileggi.

**Methodology:** Elizangela Paz de Oliveira, Amanda Flávia da Silva Rovida, Juliane Gabriele Martins, Zelinda Schemczssen-Graeff, Marcos Pileggi.

**Project administration:** Marcos Pileggi.

**Resources:** Marcos Pileggi.

**Software:** Zelinda Schemczssen-Graeff.

**Supervision:** Marcos Pileggi.

**Validation:** Elizangela Paz de Oliveira, Amanda Flávia da Silva Rovida, Juliane Gabriele Martins, Zelinda Schemczssen-Graeff, Marcos Pileggi.

**Visualization:** Elizangela Paz de Oliveira, Sônia Alvim Veiga Pileggi, Zelinda Schemczssen-Graeff, Marcos Pileggi.

**Writing – original draft:** Elizangela Paz de Oliveira, Marcos Pileggi.

**Writing – review & editing:** Elizangela Paz de Oliveira, Sônia Alvim Veiga Pileggi, Zelinda Schemczssen-Graeff, Marcos Pileggi.

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
