## [Decision Letter · Decision Letter 0]

20 Oct 2021

PONE-D-21-27759Tolerance of Pseudomonas Strain to the 2,4-D Herbicide through a Peroxidase SystemPLOS ONE

Dear authors

Thank you for submitting your manuscript to PLOS ONE. After careful consideration, we feel that it has merit but does not fully meet PLOS ONE’s publication criteria as it currently stands. Therefore, we invite you to submit a revised version of the manuscript that addresses the points raised during the review process.

We look forward to receiving your revised manuscript.

Kind regards,

Muhammad Shahid, PhD

Academic Editor

PLOS ONE

Journal Requirements:

Additional Editor Comments:

The manuscript 'Tolerance of Pseudomonas Strain to the 2,4-D Herbicide through a Peroxidase System' has been reviewed by two experts and both the reviewers raised concerns and asked the authors submit a revised version for re-consideration. Please revise the MS and provide a point-by-point response to each comment with the revised submission.

Reviewers' comments:

Reviewer's Responses to Questions

**Comments to the Author**

1. Is the manuscript technically sound, and do the data support the conclusions?

Reviewer #1: Yes

Reviewer #2: Yes

2. Has the statistical analysis been performed appropriately and rigorously? 

Reviewer #1: Yes

Reviewer #2: Yes

3. Have the authors made all data underlying the findings in their manuscript fully available?

Reviewer #1: Yes

Reviewer #2: Yes

4. Is the manuscript presented in an intelligible fashion and written in standard English?

Reviewer #1: No

Reviewer #2: Yes

5. Review Comments to the Author

Reviewer #1: Reviewer Response

The original manuscript titled ‘Tolerance of Pseudomonas Strain to the 2,4-D Herbicide through a Peroxidase System’, manuscript number ‘PONE-D-21-27759’ is an excellent research work and written in a good manner. The manuscript can be considered for publication process after meeting out the essential inclusions and improvements in the details provided. Hence, the authors are kindly requested to do the following,

1. Abstract section needs to modified and rewritten even more effectively avoiding few unnecessary explanations.

2. Provide better keywords in maximum numbers as per journal guidelines.

3. The introduction section can be presented with information of harmful effect of 2,4-D on human health, and can be improved with good research statement.

4. The results and discussion section needs to be modified. The authors can concentrate in providing more supporting discussion for the results obtained in the study. The discussion part is insufficient for understanding the results.

5. Give details on the mechanism of how Pseudomonas Strain managed to tolerate 2,4-D herbicide using Peroxidase System with your experiments.

6. The manuscript has detailed the results captured, but the effective explanations of the research results are missing; kindly update it under each respective header.

7. Conclusion section can be made even better and reachable with your obtained results.

8. The references are needed to be concentrated and aligned properly.

9. The entire alignment of manuscript content is inappropriate, it needs to be corrected.

10. Kindly correct the tenses of language used under each section and also correct the spellings.

Reviewer #2: The manuscript needs following amendments;

Line#122: DNA polymerase is always written in Units like 0.5Units.

Line#123: Kindly, mention the total reaction mixture volume.

Line#124/125: Kindly, mention thermocycler cycles with stages like Initial Denaturation at 95℃/2mins, for 40 cycles: Denaturation at 95℃/30 sec, Annealing at 57.7℃/30sec, Extension at 72℃/3min; Final-Extension at 72℃/5mins & 4℃/∞.

Line#145: Kindly, repeat the above-mentioned suggestion for this section also.

Line#212: MDA stands for?

Line#279/298/315/340/372/412/443/457: As per Journal’s Format, Figure Legends are to be mentioned at end of the text.

Line#279: Kindly, correct the figure legend font as per Journal’s Format.

Line#392: Kindly, correct “as ah response system”, as “as response system”.

6. PLOS authors have the option to publish the peer review history of their article (what does this mean?). If published, this will include your full peer review and any attached files.

Reviewer #1: No

Reviewer #2: **Yes: **Syeda Zahra Abbas

---

## [Author Response · Author response to Decision Letter 0]

4 Nov 2021

Response to Reviewers

Dear Dr. 

Muhammad Shahid, PhD

Academic Editor

PLOS ONE

Thank you for the work of the editor and reviewers in evaluating and for the considerations made to improve the submitted manuscript.

We are submitting this letter that responds to each point raised by the academic editor and reviewers, the marked-up copy of the manuscript that highlights changes made to the original version, and an unmarked version of the revised paper without tracked changes. 

Sincerely, the authors.

Reviewer #1: Reviewer Response

The original manuscript titled ‘Tolerance of Pseudomonas Strain to the 2,4-D Herbicide through a Peroxidase System’, manuscript number ‘PONE-D-21-27759’ is an excellent research work and written in a good manner. The manuscript can be considered for publication process after meeting out the essential inclusions and improvements in the details provided. Hence, the authors are kindly requested to do the following,

Authors` response: the authors thank the reviewer for the critical evaluation of the manuscript. We realize that the suggested modifications are aimed at improving the fluidity and objectivity of the text. Therefore, we incorporated them into the manuscript with the indications of the lines where these changes were inserted in the Revised Manuscript with Track Changes.

1. Abstract section needs to modified and rewritten even more effectively avoiding few unnecessary explanations.

Authors´ response: the abstract has been rewritten, located on lines 28-65. 

2. Provide better keywords in maximum numbers as per journal guidelines.

Authors´ response: The following key-words were added to PlosOne Manuscript Submission System: Oxidative stress; enzyme response; contaminated environment adaptation; bacteria response system; bacterial adaptation; phenotypic plasticity; catalase; superoxide dismutase; agriculture; herbicide tolerance; 2,4-Dichlorophenoxyacetic acid; bacterial communities; biofilm; pesticides; Pseudomonas; 2,4-D toxicity; antioxidative response system; ascorbate peroxidase; guaiacol peroxidase; hydrogen peroxide; Fe–SOD; Mn–SOD; KatA; enzyme isoforms; quorum sensing; microbiome; microbiome diversity; microbiome function; environment sustainability; human health; weeds; crop; environmental imbalances; herbicide chemical structure; chlorine; hydroxyl radical; membrane lipids; reactive oxygen species; bacterial aerobic metabolism; herbicide degradation; herbicide tolerance; bacterial growth; ribosomal 16S gene; polymerase chain reaction; DNA sequencing; cell viability; lipid peroxidation; malondialdehyde; bacterial phylogeny; stress indicators.

3. The introduction section can be presented with information of harmful effect of 2,4-D on human health, and can be improved with good research statement.

Authors´ response: a paragraph on possible effects of 2,4-D on human health has been added to the lines 87-96, with articles being inserted in the references and indicated below, and statements about this in the abstract, lines 61-62, and conclusions, to the line 552.

Rydz E, Larsen K, Peters CE. Estimating Exposure to Three Commonly Used, Potentially Carcinogenic Pesticides (Chlorolathonil, 2,4-D, and Glyphosate) Among Agricultural Workers in Canada. Ann Work Expo Health. 2021 May 3;65(4):377-389. doi: 10.1093/annweh/wxaa109. Erratum in: Ann Work Expo Health. 2021 Jul 3;65(6):740. PMID: 33336237.

Al-Amri I, Kadim IT, AlKindi A, Hamaed A, Al-Magbali R, Khalaf S, Al-Hosni K, Mabood F. Determination of residues of pesticides, anabolic steroids, antibiotics, and antibacterial compounds in meat products in Oman by liquid chromatography/mass spectrometry and enzyme-linked immunosorbent assay. Vet World. 2021 Mar;14(3):709-720. doi: 10.14202/vetworld.2021.709-720. Epub 2021 Mar 22. PMID: 33935417; PMCID: PMC8076474.

Spirhanzlova P, Fini JB, Demeneix B, Lardy-Fontan S, Vaslin-Reimann S, Lalere B, Guma N, Tindall A, Krief S. Composition and endocrine effects of water collected in the Kibale national park in Uganda. Environ Pollut. 2019 Aug;251:460-468. doi: 10.1016/j.envpol.2019.05.006. Epub 2019 May 2. PMID: 31103006.

Bianchi E, Lessing G, Brina KR, Angeli L, Andriguetti NB, Peruzzo JR, do Nascimento CA, Spilki FR, Ziulkoski AL, da Silva LB. Monitoring the Genotoxic and Cytotoxic Potential and the Presence of Pesticides and Hydrocarbons in Water of the Sinos River Basin, Southern Brazil. Arch Environ Contam Toxicol. 2017 Apr;72(3):321-334. doi: 10.1007/s00244-016-0334-0. Epub 2017 Jan 28. PMID: 28132076.

Faisal Islam, Jian Wang, Muhammad A. Farooq, Muhammad S.S. Khan, Ling Xu, Jinwen Zhu, Min Zhao, Stéphane Muños, Qing X. Li, Weijun Zhou,Potential impact of the herbicide 2,4-dichlorophenoxyacetic acid on human and ecosystems, Environment International, Volume 111, 2018, Pages 332-351, ISSN 0160-4120, https://doi.org/10.1016/j.envint.2017.10.020.

Wenjing Song, Yanjian Wan, Ying Jiang, Zhengdan Liu, Qi Wang, Urinary concentrations of 2,4-D in repeated samples from 0–7 year old healthy children in central and south China, Chemosphere, Volume 267, 2021, 129225, ISSN 0045-6535, https://doi.org/10.1016/j.chemosphere.2020.129225.

Monica K. Silver, Jie Shao, Mingyan Li, Chai Ji, Minjian Chen, Yankai Xia, Betsy Lozoff, John D. MeekePrenatal exposure to the herbicide 2,4-D is associated with deficits in auditory processing during infancy, Environmental Research, Volume 172, 2019, Pages 486-494, ISSN 0013-9351, https://doi.org/10.1016/j.envres.2019.02.046.

4. The results and discussion section needs to be modified. The authors can concentrate in providing more supporting discussion for the results obtained in the study. The discussion part is insufficient for understanding the results.

Authors´ response: the discussion was expanded into different topics in the article, such as in the lines 318-322; 336-340; 367; 473-477; 523-528; 545; as well as the respective references were inserted:

Aguiar LM, Dos Santos JB, Barroso GM, Laia ML, Gonçalves JF, da Costa VAM, Brito LA. Influence of 2,4-D residues on the soil microbial community and growth of tree species. Int J Phytoremediation. 2020;22(1):69-77. doi: 10.1080/15226514.2019.1644289. Epub 2019 Jul 25. PMID: 31342787.

Yang Z, Xu X, Dai M, Wang L, Shi X, Guo R. Rapid degradation of 2,4-dichlorophenoxyacetic acid facilitated by acetate under methanogenic condition. Bioresour Technol. 2017 May;232:146-151. doi: 10.1016/j.biortech.2017.01.069. Epub 2017 Feb 11. PMID: 28219052.

Freitas PNN, Rovida AFDS, Silva CR, Pileggi SAV, Olchanheski LR, Pileggi M. Specific quorum sensing molecules are possibly associated with responses to herbicide toxicity in a Pseudomonas strain. Environ Pollut. 2021 Nov 15;289:117896. doi: 10.1016/j.envpol.2021.117896. Epub 2021 Aug 2. PMID: 34358867.

Tang J, Hu Q, Lei D, Wu M, Zeng C, Zhang Q. Characterization of deltamethrin degradation and metabolic pathway by co-culture of Acinetobacter junii LH-1-1 and Klebsiella pneumoniae BPBA052. AMB Express. 2020 Jun 3;10(1):106. doi: 10.1186/s13568-020-01043-1. PMID: 32495133; PMCID: PMC7270285.

Hassett DJ, Ma JF, Elkins JG, McDermott TR, Ochsner UA, West SE, Huang CT, Fredericks J, Burnett S, Stewart PS, McFeters G, Passador L, Iglewski BH. Quorum sensing in Pseudomonas aeruginosa controls expression of catalase and superoxide dismutase genes and mediates biofilm susceptibility to hydrogen peroxide. Mol Microbiol. 1999 Dec;34(5):1082-93. doi: 10.1046/j.1365-2958.1999.01672.x. PMID: 10594832.

5. Give details on the mechanism of how Pseudomonas Strain managed to tolerate 2,4-D herbicide using Peroxidase System with your experiments.

Authors´ response: a proposal for a 2,4-D tolerance mechanism by the Pseudomonas strain is presented in lines 473-477; 523-528, Abstract and Conclusions sections.

6. The manuscript has detailed the results captured, but the effective explanations of the research results are missing; kindly update it under each respective header.

Authors´ response: the discussion was expanded into different topics in the article, such as in the lines 318-322; 336-340; 367; 473-477; 523-528; 545, as well as the respective references, already described, were inserted.

6. Conclusion section can be made even better and reachable with your obtained results.

Authors´ response: the conclusions were modified to include the reviewer's suggestions in the lines 567, 570, 572 and 576-577. 

7. The references are needed to be concentrated and aligned properly.

Authors´ response: the reviewer's suggestions were followed.

8. The entire alignment of manuscript content is inappropriate, it needs to be corrected.

Authors´ response: the reviewer's suggestions were followed.

10. Kindly correct the tenses of language used under each section and also correct the spellings.

Authors´ response: a new language review was made, as suggested by the reviewer.

Reviewer #2: The manuscript needs following amendments;

Line#122: DNA polymerase is always written in Units like 0.5Units.

Authors´ response: the authors thank the reviewer for the critical evaluation of the manuscript. The text has been changed, as suggested by the reviewer.

Line#123: Kindly, mention the total reaction mixture volume.

Authors´ response: the text has been changed, as suggested by the reviewer.

Line#124/125: Kindly, mention thermocycler cycles with stages like Initial Denaturation at 95℃/2mins, for 40 cycles: Denaturation at 95℃/30 sec, Annealing at 57.7℃/30sec, Extension at 72℃/3min; Final-Extension at 72℃/5mins & 4℃/∞.

Authors´ response: the text has been changed, as suggested by the reviewer, and following the International System of Units.

Line#145: Kindly, repeat the above-mentioned suggestion for this section also.

Authors´ response: the text has been changed, as suggested by the reviewer, and following the International System of Units.

Line#212: MDA stands for?

Authors´ response: the abbreviation was explained in the text, as suggested by the reviewer.

Line#279/298/315/340/372/412/443/457: As per Journal’s Format, Figure Legends are to be mentioned at end of the text.

Authors´ response: I understand the reviewer's suggestions, but the journal's rules are as follows:

Figure captions

Figure captions must be inserted in the text of the manuscript, immediately following the paragraph in which the figure is first cited (read order). Do not include captions as part of the figure files themselves or submit them in a separate document.

Line#279: Kindly, correct the figure legend font as per Journal’s Format.

Authors´ response: the formatting of the titles of the figures were modified, as suggested by the reviewer.

Line#392: Kindly, correct “as ah response system”, as “as response system”.

Authors´ response: the text has been changed as suggested by the reviewer.

---

## [Editor Report · Decision Letter 1]

17 Nov 2021

Tolerance of Pseudomonas Strain to the 2,4-D Herbicide through a Peroxidase System

PONE-D-21-27759R1

Dear Dr. Marcos Pileggi

We’re pleased to inform you that your manuscript has been judged scientifically suitable for publication and will be formally accepted for publication once it meets all outstanding technical requirements.

Kind regards,

Muhammad Shahid, PhD

Academic Editor

PLOS ONE
---

## [Editor Report · Acceptance letter]

23 Nov 2021

PONE-D-21-27759R1 

Tolerance of *Pseudomonas* Strain to the 2,4-D Herbicide through a Peroxidase System 

Dear Dr. Pileggi:

I'm pleased to inform you that your manuscript has been deemed suitable for publication in PLOS ONE. Congratulations! Your manuscript is now with our production department. 

Kind regards, 

on behalf of

Dr. Muhammad Shahid 

Academic Editor

PLOS ONE